# The Winding Road from Origin to Emergence (of Life)

**DOI:** 10.3390/life14050607

**Published:** 2024-05-09

**Authors:** Wolfgang Nitschke, Orion Farr, Nil Gaudu, Chloé Truong, François Guyot, Michael J. Russell, Simon Duval

**Affiliations:** 1BIP (UMR 7281), CNRS, Aix-Marseille-University, 13009 Marseille, France; orion.farr@etu.univ-amu.fr (O.F.); ngaudu@imm.cnrs.fr (N.G.); ctruong@imm.cnrs.fr (C.T.); sduval@imm.cnrs.fr (S.D.); 2CINaM, CNRS, Aix-Marseille-University, 13009 Marseille, France; 3IMPMC (UMR 7590), CNRS, Sorbonne University, 75005 Paris, France; francois.guyot@mnhn.fr; 4Dipartimento di Chimica, Università degli Studi di Torino, 10124 Torino, Italy; michaeljrussell80@gmail.com

**Keywords:** emergence of life, 2nd law of thermodynamics, far-from-equilibrium thermodynamics, chemiosmosis, CHONPS, vitalism, primordial soup

## Abstract

Humanity’s strive to understand why and how life appeared on planet Earth dates back to prehistoric times. At the beginning of the 19th century, empirical biology started to tackle this question yielding both Charles Darwin’s Theory of Evolution and the paradigm that the crucial trigger putting life on its tracks was the appearance of organic molecules. In parallel to these developments in the biological sciences, physics and physical chemistry saw the fundamental laws of thermodynamics being unraveled. Towards the end of the 19th century and during the first half of the 20th century, the tensions between thermodynamics and the “organic-molecules-paradigm” became increasingly difficult to ignore, culminating in Erwin Schrödinger’s 1944 formulation of a thermodynamics-compliant vision of life and, consequently, the prerequisites for its appearance. We will first review the major milestones over the last 200 years in the biological and the physical sciences, relevant to making sense of life and its origins and then discuss the more recent reappraisal of the relative importance of metal ions vs. organic molecules in performing the essential processes of a living cell. Based on this reassessment and the modern understanding of biological free energy conversion (aka bioenergetics), we consider that scenarios wherein life emerges from an abiotic chemiosmotic process are both thermodynamics-compliant and the most parsimonious proposed so far.

## 1. The Prevailing Paradigm: It’s the Organics, Stupid!

Judging from the communications of major research agencies all over the world which directly or indirectly tackle the question of life’s origins, the most fundamental prerequisite for life to appear has been identified: the presence of organic molecules such as amino acids, nucleobases, lipids and sugars. The actual open questions pertain to the precise synthesis pathways for these organics and the processes which then have allowed them to evolve into living cells. The introduction to NASA’s Astrobiology Strategy 2015 roadmap for example reads: “A major goal of research on this topic in astrobiology is to understand how the abiotic (non-biological) production of small molecules led to the production of large and more complex molecules, prebiotic chemistry, and the origin of life on Earth” [1]. Obedient to this instruction, all recent space missions had an eye on detecting organics, the “seeds of life”. They intend to “understand the role asteroids may have played in delivering life-forming compounds to Earth” [2], “to investigate questions … where the organic matter that forms life originally came from” [3] and thus to “grant us direct access to some of the ingredients that likely ended up in the prebiotic soup that eventually resulted in the origin of life on Earth” [4]. This focus on the detection of organic molecules in the solar system appears natural if “we begin with the premise that life emerged on Earth from chemistry that led to the synthesis of molecular building blocks, which, in turn, self-assembled to form cells” [5].

The above listed citations of course resonate with the conviction expressed more than 150 years ago by Charles Darwin in his oft-cited correspondence with Joseph Dalton Hooker [6]: “But if (and oh what a big if) we could conceive in some warm little pond …, that *a protein compound was chemically formed, ready to undergo still more complex changes*, …” (emphasis added by the authors). As we will see below, Darwin’s assurance that it is the chemically formed “protein compounds” that would “readily” (that is, spontaneously) transform into more complex structures and eventually bring forth life, is deeply rooted in and informed by the scientific/intellectual background of his century. As evidenced by the citations above, Darwin’s conviction is as unswerving today as it was in his days, despite many scientific paradigms having substantially changed at the turn from the 19th to the 20th century and even more so during the 2nd half of the 20th century, as we will discuss in this article. While, as we will argue, Darwin has merely expressed the scientific paradigm of his days rather than having put profound intellectual efforts into coming up with this clause in a letter to a colleague, the search for the “chemically formed protein compounds” nevertheless turned into the overarching roadmap (up to the present day) for understanding the appearance of life on our planet. The decisive big leap forward following the direction suggested by this roadmap obviously was Stanley Miller’s demonstration that organic molecules (mainly amino acids) are generated if suitable mixtures of inorganic gases (H_2_, CO_2_, CH_4_) and solutes (mainly ammonia in water) were exposed to electrical discharges [7,8]. Several years later, the realization that the initial conditions underlying the Miller/Urey experiments were inappropriate for the early Earth (assumed by geologists a century before) sparked the search for other pre-biotic synthesis pathways for organics (preferably amino acids, nucleobases and lipids, the predominant ingredients of extant life on Earth). 70 years of post-Miller/Urey efforts to uncover such pathways have led to a veritable zoo of chemical reaction schemes applicable to a large number of putative initial environmental conditions on the primordial planet and beyond (Figure 1). Most of these generally follow the tracks of the Miller/Urey experiment in that they involve standard chemical reactions under Earth-relevant conditions [5,9,10] while a few appeal to more exotic schemes such as involving high-energy protons from Coronal Mass Ejection (CME) events [11] or ionizing radiation from natural nuclear reactors [12]. However, organic molecules can obviously also be formed under even more “unearthly” conditions such as on comets, asteroids or in the interstellar medium as evidenced by the space missions mentioned above. For example, certain amino acids have been detected on comets [13], the nucleobases used by extant life have been found in meteorites [14] and in samples returned to Earth from asteroid Ryugu during JAXA’s Hayabusa-2 mission [15,16] and polycyclic aromatic hydrocarbons found on Ryugu were identified as having been formed in the interstellar medium and further processed on the asteroid [17].

There therefore is no shortage of potential pre-biotic synthesis pathways for organic molecules and the predominant argument in origin-of-life research presently focuses on which of these pathways is the most pertinent one to the actual birth of life on our planet and whether these organics were produced in-situ (i.e., on Earth) in the atmosphere, the hydrosphere or the lithosphere, or whether they were delivered from space (Figure 1).

This tacitly eschews explaining why organic molecules would spontaneously “evolve” into ever-complexifying systems, finally reaching a state which can be recognized as “living”, while no other type of matter seemed able to pull off this feat. The narrative of “organic molecules engendering life” has become so deeply ingrained in the mindset of both scholars and the general public during the second half of the 20th century that no questioning seemed appropriate. Incidentally, the same appears to be the case for the topic “life” in general, where immense progress in microbiology, biochemistry and genetics since the 1950s uncovered the “hows” of life’s mechanisms obscuring the question of why life does what it does. In striking contrast, from the early 1800s into the early 20th century, scholars were painfully aware of life’s apparent antagonism to the behaviour of the physical world.

## 2. Physics vs. Biology; from Sadi Carnot to Erwin Schrödinger and from Jöns Jacob Berzelius to Stanley Miller

These were the times when thermodynamics was elaborated, morphing from Sadi Carnot’s considerations of the efficiency limits of heat engines in 1824 through Rudolf Clausius’ and William Thomson’s empirical formulations of the 2nd law of thermodynamics in the middle of the 19th century into the statistical rationalization of the 2nd law initiated by J.W. Gibbs and definitively conceptualized by Ludwig Boltzmann during the 2nd half of the 19th century (for a timeline, see Figure 2). One of the major take-home messages first formulated by Thomson and later further developed by von Helmholtz and William Rankine was the notion of the ultimate “heat death” of the universe. The 2nd law of thermodynamics indeed predicts an inevitable degradation over time of the free energy available to do work, a conclusion later rationalized by trajectories of physical systems from less to more probable states in Boltzmann’s statistical mechanics. 19th century scholars therefore were fully aware that physical systems tend towards increasing disorder while, in striking contrast, life seemed to do exactly the opposite. The antithetical behaviours of the physical world on one side and life on the other obliged 19th century biologists to come up with novel hypotheses attempting to resolve this cognitive dissonance.

### 2.1. The Century of Vitalism (~1800 to ~1900)

It seems counterintuitive from our present perspective that the year 1859 saw both the publication of Louis Pasteur’s refutation of spontaneous generation and Charles Darwin’s “On the *origin* of *species*”. Pasteur’s brilliant series of experiments severely challenged the millennia-old notion, famously expressed by Aristotle as early as the 4th century BCE (“So with animals, some spring from parent animals according to their kind, whilst others grow spontaneously and not from kindred stock; and of these instances of spontaneous generation some come from putrefying earth or vegetable matter, as is the case with a number of insects, while others are spontaneously generated in the inside of animals out of the secretions of their several organs.” [23]), that life can spontaneously emerge from inanimate matter. Thus Pasteur held that life could only derive from life while Darwin’s theory of evolution saw the extant diversity and complexity of ecosystems emerge from ever simpler organisms finally arriving at a point in history when life rose out of an inanimate world. In our days, Pasteur’s and Darwin’s views appear to represent irreconcilable opposites. Not so in the 19th century! Nascent organic chemistry in the late 1800s and early 1900s was strongly influenced by (but also intensely struggling with) the concept of “vitalism”. Chemists in those days seemingly observed an insurmountable barrier between inorganic and organic molecules, the latter considered to only being produced by living organisms. This led to the notion that organic molecules must be endowed with some ill-defined “vital force” which enables them to bring forth life and, vice versa, require organisms for their synthesis. Jöns Jacob Berzelius, often cited as the father of organic chemistry at the beginning of the 19th century, pleaded the cause of vitalism for the major part of his scientific life, and continued to do so even after Friedrich Wöhler, one of his former students, had shown in 1828 that the organic molecule urea could be synthesized from inorganic precursors (Figure 2). Pasteur himself considered that the experiments disproving spontaneous generation provided strong support for vitalism. Although many scientists (such as Justus von Liebig and Berzelius himself) felt intense unease concerning the “mystical” aspects of the vital force, they nevertheless stuck to vitalism for lack of a better rationalization of life’s perceived anti-thermodynamic behaviour. Darwin’s idea that “protein compounds” might be synthesized from inorganic sources, just like Wöhler’s urea, and then spontaneously further evolve into living matter thanks to the vital force inherent to organics therefore was all but outrageous within the mindset of vitalism which strongly influenced chemistry’s and biology’s thinking during the major part of the 19th century.

The development of an atomistic understanding of molecules, initiated by the concept of chemical structure proposed independently by Friedrich August Kekulé and Archibald Scott Couper in 1859 and culminating in the discovery of the electron and its role in chemical bonding at the turn from the 19th to the 20th century, did away with the fundamental dichotomy between inorganic and organic chemistry. Organic molecules were henceforth “just molecules” removing any scientific basis for assigning special life-inducing properties to hydrocarbons, as complex as they may come. Vitalism became increasingly disreputable, although a handful of biologists subscribed to vitalist principles well into the first decades of the 20th century [24,25,26,27]. Ernst Mayr finally buried vitalism as a biological hypothesis by declaring that “no biologist alive today would want to be classified as a vitalist” [28].

### 2.2. Organicism and Life’s Origin as a One-Off Event

The vital force of organics having become obsolete, a few scientists trying to rationalize life’s whereabouts turned to “Organicism” and the idea that the “organic whole” of specific assemblies of hydrocarbons must possess properties which cannot be explained by reductionist analysis of their constituents [29]. J.B.S. Haldane, together with Aleksandr Oparin considered as the fathers of the “prebiotic-soup-hypothesis” for the origin of life, i.e., the conceptual framework leading to the Miller/Urey experiments, strongly built his views of life and its origin on organicism. Shifting the life-enabling properties from the building blocks, the organic molecules, towards the “organismal whole” seemingly did away with the requirement for vitalist concepts since abiotically produced organics now “only” needed to accidentally assemble into first organisms which then allowed the “organismal” properties to kick in. Such accidental order-increasing (anti-entropic) trajectories were indeed conceivable in the framework of Boltzmann’s statistical interpretation of thermodynamics. Just as the probability-driven mixing of two distinguishable, previously compartmentalized, gases has a non-zero probability to, over time, transiently go through the unmixed state again, random mixtures of organic molecules might well accidentally form more ordered assemblies. Still, two comments impose themselves here:(i)According to statistical thermodynamics, the probability of fluctuation-induced increase in order decreases steeply with complexity of these systems and the degree of complexity required to arrive at even the simplest imaginable cellular structures is doubtlessly enormous. A number of biologists around the middle of the 20th century were perfectly aware of the vanishingly small probability for life to arise this way by mere chance prompting for example Jacques Monod to proclaim that “… man at last knows that he is alone in the unfeeling immensity of the universe, out of which he has emerged only by chance” [30]. It is in the framework of Monod’s thinking that the term “origin”, as describing life’s coming into being out of a soup of organics, unveils its profound sense: a freak event which obviously did happen (since we know for sure that there is life on Earth) but which was so unlikely that there is no point expecting it to have happened elsewhere in the cosmos. Needless to say that many citations listed in Section 1, while deeply rooted in the prebiotic soup paradigm, nonchalantly ignore the inevitability of Monod’s logic!(ii)To the post-19th-century biologists, the notion of anti-entropic properties inherent in organismal assemblies of organics may have been intuitively more acceptable than the idea of a vitalist force in hydrocarbons. We do observe this anti-entropic tendency in all living things, don’t we? Still, the order-generating properties of organismal assemblies as envisaged by the organicists are no less mystical then the vital force and relying on such properties takes the phenomenon “life” out of the explanatory framework of the physical sciences. In 1944, Erwin Schrödinger’s book “What is Life?” finally put an end to speculations about life’s specialness by rigorously treating life as part of the physical world [18].

## 3. The Demystification of Life and Its Integration into the Physical Sciences

### 3.1. Schrödinger and the Crux of the 2nd Law

As Schrödinger pointed out, the main misconception dooming all efforts to understand life and its beginnings since the days of the early vitalists lay in considering living things as independent entities, that is, as thermodynamically closed systems. That this notion is fatally misguided will be brought home to anybody trying to escape Earth’s atmosphere without taking the proper supply of molecular oxygen. We “burn” the food that we eat with the oxygen we breathe and it is the free energy available in this reaction which allows us to live. This is true for all life on Earth and only the sources of reductants and oxidants involved in this “life-enabling” reaction vary between species [31]. To Schrödinger-the-physicist, this fact was a trivial but also inescapable consequence of the 2nd law of thermodynamics. Entropy decrease in living beings necessarily implies (a) that they are part of a larger system which itself is in a state of thermodynamic disequilibrium and (b) that order-generation within the living system is only allowed by the 2nd law if it takes the larger system closer to equilibrium. In Schrödinger’s words: Life has to “eat neg-entropy” [18]. Unsurprisingly, biologists were unable to find even a single exception to the rule that life requires the input of free energy from the environment in order to exist. Schrödinger’s considerations therefore have made the existence of living beings compliant with the laws of the physical world. However, while we now know that environmental free energy rather than a mystical vital force or organismal hat-tricks allows the generation of order by living systems, why does life behave this way while abiotic systems don’t? Or do they?

### 3.2. Non-Living Systems Can Display Entropy-Decreasing Features; the End of the Life/Non-Life-Dichotomy

The world of physical phenomena is indeed replete with locally entropy-decreasing processes. Such processes are observed from fluid dynamics (e.g., turbulence above a critical Reynolds number, Taylor-Couette flows or Benard cells) through atmospheric systems (from Kelvin-Helmholtz instabilities to large scale cyclones) to chemical reaction schemes (for example the iconic Belouzov-Zhabotinsky reaction). The common feature of all such phenomena is that they arise within larger systems far from thermodynamic equilibrium and that their local entropy decrease is constantly fed by the flux of free energy taking the larger system closer to equilibrium.

This conversion of free energy (i.e., low entropy) of the larger system into entropy decrease (ordering) in the subsystem ensures nominal compliance with the 2nd law but per se doesn’t explain how this conversion is accomplished. Progress in the theoretical description of dynamic trajectories of systems far from thermodynamic equilibrium over the last 50 years [19] together with significantly improved capabilities of numerical simulations now provide an in-depth understanding of how the respective systems under specific circumstances occasionally decrease entropy, i.e., generate spatially and/or temporally ordered states [20].

The term “dissipative structures”, coined by Ilya Prigogine in the 1970s (Figure 2) to describe such states, emphasizes both the structuring of the subsystems and the free energy dissipation in the larger, embedding ones. The appearance of a dissipative structure therefore corresponds to an “emergence”, that is, a phenomenon not inherent in the reductionist components of a system but a result of the thermodynamic forces driving the system’s trajectory [19]. The emergence of a dissipative structure therefore is contingent on the free energy environment driving the system’s dynamics and not on what the system is made of!

Erwin Schrödinger showed that life, to comply with the 2nd law, must necessarily be part of a larger system (the living entities plus their environment) which itself is in thermodynamic disequilibrium. It is therefore inescapable to recognize life as an emergent dissipative structure following the same general principles as the physical/chemical systems described above. In this framework of thinking, the fact that life appeared on planet Earth is first and foremost due to the thermodynamic circumstances at the time and place of its birth and not to the reductionist properties of its constituents. The somewhat mystical vital forces of organic molecules or the supposed order-generating features of organismal assemblies finally find their rationalization in the tendency of systems which are far from thermodynamic equilibrium to form dissipative structures. Rather than life having “originated” as a thermodynamically extremely unlikely event, modern understanding of far-from-equilibrium thermodynamics suggests that we better refer to this phenomenon as a thermodynamically-driven “emergence of life” as proposed previously [32,33].

However, does this change of paradigm really deprive organic molecules of their central importance in this process? Surely the environmental thermodynamic disequilibria driving the emergence of life must have been exploited by organic-molecule-based mechanisms, just as they are generally considered to be in extant life?

## 4. Reassessing the Organo-Centric View of Living Entities

### 4.1. CHONPS vs. FeNiMoWCo+; Giant vs. Dwarf?

The conviction that life is all about organic molecules is related to the even more fundamental notion that the prerequisites for life on Earth come down to 6 relatively light elements, affectionately called CHONPS (some authors prefer the easier to pronounce term SPONCH). “Life as we know it is based on six elements, carbon (C), hydrogen (H), oxygen (O), nitrogen (N), phosphorous (P), and sulphur (S)—typically abbreviated as CHONPS” [34], or “A living cell consists of two basic kinds of polymers … composed of six elements abbreviated CHONPS” [35]. Not only are the CHONPSes considered the compositional bricks of life, they also seem to be responsible for life’s defining processes, since “Biochemistry on Earth makes use of the key elements … (or CHONPS)” [36]. Indeed, per dry weight of living matter, 96% are organic molecules and 3% are light-atomic-weight ions such as sodium, potassium, chloride etc. It therefore is unsurprising that the early organic chemists in the times of Berzelius found only carbon (~50%) with small admixtures of oxygen (20–30%), nitrogen (~15%) and even lower amounts of hydrogen, phosphorous and sulphur [37].

We now know that in addition to the CHONPSes, further elements, mainly consisting of transition metals, make up roughly 1% of the dry weight of prokaryotic cells (Figure 3a). This marginal fraction of mass is widely taken to indicate a correspondingly marginal role in life in general and in particular in its appearance on the early Earth. However, vital roles of some of these elements, later to become the “essential trace elements”, have been recognized almost two centuries ago. Such essential trace elements were first observed in the middle of the 19th century when Jules Raulin, a student of Pasteur’s, described that growth of *Aspergillus niger* fully depended on the presence of zinc. By the beginning of the 20th century, microbiologists had worked out a catalog already comprising the majority of the essential trace elements identified today. We therefore know since more than a hundred years that life is not only based on the CHONPSes but also on a number of further elements such as Mg, Mn, Fe, Cu, Zn, Ni, Mo, W, Co, V and so on (summarily referred to in this work as FeNiMoWCo+ (A host of these heavier atoms play crucial roles in metabolic processes of extant life. We have chosen to call them “FeNiMoWCo+” since iron, nickel, molybdenum/tungsten and cobalt have specifically been put forward as essential agents in putatively ancient reaction schemes, such as H2-oxidation, CO2-reduction, CH4-oxidation, NO3^−^-reduction etc. [31,38,39]. The “+”-sign allows for further transition metals, the importance of which may not yet be fully appreciated. Copper does not enter the list since it almost certainly was not bioavailable under the anoxic conditions of the primordial ocean [40]. Manganese admittedly plays a paramount role in oxygenic photosynthesis. However, as argued in the past, we do not consider photosynthesis (oxygenic or anoxygenic) to have been operating at the time of life’s emergence [31]. Incidentally, the proposed list makes for a “pronounceable” word to compete with the CHONPSes)). But surely these elements must be of lesser importance to life’s processes since they are so rare as compared to the CHONPSes, earning them the attribute “trace”? This reasoning commits a serious mistake: the biological units actually capable of “doing something useful”, such as for example converting environmental free energy into the internal (spatial and temporal) order of a living cell, are of course not individual CHONPS-atoms but the enzymes. A single enzyme, however, is made up from an enormous number of CHONPS-atoms! Middle-of-the-road prokaryotic cells (such as *E. coli*) contain slightly more than 2 million proteins (not all of which are enzymes, i.e., catalyze reactions or convert disequilibria) [41]. This same cell contains about 0.2% (by dry weight) of iron, certainly a ridiculously low quantity if compared to the close to 50% carbon or 20–30% oxygen. However, recalculating the weight percentage into actual number of Fe-ions per cell yields the staggering value of roughly 4 million. *E. coli* therefore contains more iron atoms than proteins! Even considering (i) that part of these irons are probably stored in ferritins for later use, (ii) that many enzymes actually harbour multi-iron metal clusters and (iii) that not all enzymes are metalloproteins, still yields a situation where a substantial fraction of all enzymes are indeed iron-containing metalloenzymes. Among the trace elements, iron might be considered borderline since it is required in much higher amounts than other transition metals (Fe alone accounts for one fifth of the total dry weight of all non-CHONPS-elements). However, even for the very low-abundancy trace metal molybdenum [42], numbers of atoms per cell reach about 5000, which is roughly commensurate with the number of individual bioenergetic electron transfer chains when *E. coli* grows by (anaerobic) respiration of nitrate. The comparison in Figure 3 illustrates that the picture of metals as marginal components in a living cell is fatally flawed.

### 4.2. CHONPS vs. FeNiMoWCo+; Which One Does the Work?

To the bioenergetics community, i.e., the field studying the molecular mechanisms performing free energy conversion, all this comes as no surprise since more than 50 years of research have firmly established that it is the metals (Fe, Cu, Ni, Mo, …) that do the job and that the protein scaffolds holding the metal centres mainly serve to fine-tune the physico-chemical properties of these inorganic clusters. Rather than being marginal phenomena in life’s processes, metal ions are the dominant players! This evidence is further brought home by the observations that roughly half of all enzymes are metalloenzymes [43,44] and that there does not seem to be a single recognized metabolic pathway that doesn’t contain at least one metalloenzyme [45].

There likely is one subgroup of the Origin-of-Life community which will remain unfazed by the above described considerations, that is, RNA-world proponents. In their framework of thinking, (metallo-)protein-based processes, certainly the major mechanisms in extant life, did not operate at life’s beginnings. Rather, it were catalytic RNA-based molecules (so-called ribozymes) which in the beginning performed these tasks, later taken over by proteinaceous enzymes. While we will here refrain from discussing the pros and cons of the RNA-world hypothesis, we cannot help noticing that all naturally occurring ribozymes (whereof there aren’t a great many anyway) are in fact metallo-ribozymes depending on metal ions for catalysis and/or 3D-structuring [46].

Our present understanding of both the quantities and the functional roles of metal ions in living cells therefore fully justifies the opening statement in a recent review article on the young field of metallomics: “Without metal ions life is not possible and would not have evolved in the first place” [47]. This statement obviously is a far cry from Darwin’s idea that protein compounds would readily evolve towards living entities.

This paramount role of metal ions in enabling life should not come as a surprise. Metal-free peptides (i.e., chains of amino acids) have no a priori catalytic activities, largely due to the fact that these activities require specific structural foldings—foldings which moreover often depend, for both form and function, on binding specific metal ion complexes (thereby conferring a more-or-less stringent constraint on the amino acid sequence forming the fold). In this general picture, strikingly, one class of metabolic processes, whose mediation almost invariably requires these metal-ion enabled protein fold structures, stands above all others in the affairs of life. Indeed, as we will discuss below, the most important processes in metabolism, that is, the mechanism underpinning the possibility for life to exist according to the 2nd law, are redox reactions. In the overwhelming majority of cases, redox-active enzymes are metalloenzymes and it is the metal ion (or clusters thereof) that does the job. Only in very rare exceptions (e.g., cytochrome oxidase or photosystem II), protein-integral amino acids participate in redox reactions and in all these cases, the parent enzymes are multi-metal-ion systems with the respective amino acids performing isolated electron transfer steps within overall metal-based chains.

Ribozymes, strictly speaking also metalloenzymes as pointed out above, have only extremely specific catalytic properties, mainly directed towards themselves, such as auto-splicing, or are involved in the replicatory functions of the ribosome [48]. None of these catalytic functions is even only remotely related to the life-enabling process of metabolism.

In most biochemistry textbooks, the existence of metalloenzymes is treated as Nature having cleverly realized that catalytic performance of an enzyme can be improved by recruiting metal ions in specific binding sites. This way of thinking is both true to the “organics-first” paradigm and likely influenced by the way technology (in particular in the chemical industry) operates. Kinetically sluggish reaction schemes (often requiring cost-intensive high temperatures and high pressures) are frequently substantially accelerated by the additions of certain metals. A number of arguments, however, speak against the “from metal-free enzymes to metalloproteins”-notion. (i) All apo-metalloenzymes, i.e., deprived of their metal centres, involved in energy converting metabolism aren’t less active than their holo-counterparts but simply dead! (ii) By contrast, their protein-free metal-clusters, frequently encountered in certain minerals [38], often perform the same reactions as when integrated in polypeptides [49]. (iii) In the case of environmental scarcity of certain metals, life has found ways to use other available metals as replacement. For example, in iron-depleted habitats, the canonical tasks of Fe-containing enzymes are routinely taken over by structurally (and sequence-wise) fully unrelated copper-enzymes as exemplified by the *cd*1-Nir to Cu-Nir (both enzymes reducing nitrite to nitric oxide) transition. If pre-existing metal-free life would have recruited the metal-ions, one would certainly have to assume that we should still be able to find such organisms in environments highly depleted in all metal ions. This does not seem to be the case. To the best of our knowledge, no species has been observed so far which can live without metals. This is even true for species only deduced from metagenome analyses, where routinely a full host of metalloenzymes are identified via their tell-tale specific cofactor-binding sequences. Both in the environment and under laboratory conditions, truly metal-free media are simply sterile! We therefore consider it an inescapable fact that FeNiMoWCo+ vastly surpasses the importance of CHONPS in rationalizing why life on Earth was able to emerge.

## 5. A 2nd Law-Compliant Emergence of Life; from Generalities to Specifics

### 5.1. The 2nd Law Meets Prebiotic Chemistry

Almost 80 years after the publication of Schrödinger’s book, the thermodynamic argument begins to make occasional inroads into the organic soup fortress which previously had managed to remain hermetic to criticism over the 2nd half of the 20th century (and criticism there was … [50,51,52,53,54]). NASA’s website describing research on putative life on Mars for example reads: “… the origin of life requires the presence of carbon-based molecules, liquid water *and an energy source*” (emphasis by the authors) [55]. Tellingly, the nature of this “energy” remains unspecified. Even more worrisome is the fact that the energies put forward in many articles are of the sort of photons, ionizing radiation, heat or electrical discharge (a combination of heat and reducing power). Such “energies” can indeed assist in overcoming activation barriers of specific reactions taking the reactant/product couples more rapidly to their thermodynamic equilibria. This is what happens when small amounts of organics are detected after exposing high amounts of precursor molecules to UV-irradiation, bombardment with all sorts of ionizing radiations or electrical discharge. However, this is clearly not what life does! Life takes the precursors and produces extremely out-of-equilibrium concentrations of the products. To be able to do this, these processes must by driven by disequilibria in life’s environment, i.e., by “free energy” in the sense of Gibbs’ formulation of the 2nd law, as discussed above [21,22,56,57,58]. The crucial distinction between energy (a strictly conserved entity as enshrined in the 1st law of thermodynamics) and non-conserved “free energy” (in fact not an energy at all but a measure of how far from equilibrium a system is and thus of its ability to do work) has been pointed out repeatedly in the past [58,59,60]. For example, it isn’t heat (a 1st law type energy) but a temperature gradient (a 2nd law type free energy, that is, a thermodynamic disequilibrium) that produces hurricanes. Of course, energies can be transformed into Gibbs free energy, such as when for example radioactive decay in Earth’s mantle generates heat and hence convection which eventually will via the process of serpentinization bring about redox disequilibria [61]. However, the Gibbs free energy potentially being converted into other disequilibria, e.g., life’s local decrease in entropy, is this ultimate redox disequilibrium and not the (1st law-type) energy of radioactive decay.

Leaving this (major) detail aside, a growing number of articles admittedly now acknowledge Schrödinger’s stricture that the emergence of life must obey the 2nd law [62,63,64,65,66]. The necessity of pushing the subsystem life away from thermodynamic equilibrium is now discussed even in articles from the prebiotic soup community. For example, a consecutive sequence of disequilibrium-maintaining inputs of energy (some of which are 1st law-type energies rather than Gibbs free energies, see above) are proposed to take prebiotic organics step-by-step up to ever more extreme disequilibria and eventually to cellular life which, at supposedly “higher stages of evolution” [63] converts free energy through a process the biologists call “chemiosmosis” [63].

### 5.2. Extant Life’s Universal Steam Engine: Chemiosmosis

The mechanism of chemiosmosis (we will explain the process in more detail below) is common to all life on planet Earth and is therefore widely considered to be the way by which the so-called Last Universal Common Ancestor (LUCA) harvested environmental free energy [31,67,68,69,70,71]. It took the biological sciences more than half a century to unravel the major mechanisms of the chemiosmotic system and several key processes are still not well understood. Chemiosmotic enzymes are among the most complex molecular machineries involved in cellular processes [72,73] and it is therefore only natural that evolutionary biologists generally regard chemiosmosis as a highly evolved mechanism, despite its ubiquity in extant life and likely deep ancestry going back to the common ancestor of Bacteria and Archaea. Chemiosmosis’ deep evolutionary roots are further emphasized by the observation that its common principle of action (see below) has been retained since the times of the LUCA while adapting to a bewildering variety of sources of environmental free energies (all of which ultimately come down to electrochemical free energy, see [74]) allowing life to spread to virtually every habitat on the planet providing redox disequilibria of some sorts. At later stages of evolution, photosynthetic organisms managed to generate such redox disequilibria via charge separation induced by visible light photons while still fully relying on the chemiosmotic principle [31,74]. Even the plethora of different enzyme systems converting these redox disequilibria into life’s spatial and temporal ordering have now been shown to be built up from a very small number of basic metalloprotein domains, many of which can be phylogenetically traced back to the LUCA [75,76,77,78,79,80]. The chemiosmotic principle for the conversion of environmental free energy into the entropy decrease characterizing life from the LUCA to extant life is therefore now firmly established.

So what does this ubiquitous process actually do? The ultimate result of the chemiosmotic mechanism is the conversion of redox disequilibria, i.e., of the simultaneous presence of sufficiently reducing and sufficiently oxidizing redox compounds into a disequilibrium ratio of polyphosphates to monomeric phosphates (where the “sufficiently” indicates that the flow of electrons from reductant to oxidant is exergonic even taking into account the free energy required to generate the phosphate/polyphosphate disequilibrium). In most metabolic processes, these polyphosphates are linked to an adenosine moiety, yielding the bioenergetically prominent molecules adenosine mono-, di- and triphosphate (AMP, ADP and ATP). In thermodynamic equilibrium at standard conditions (1 atmosphere, room temperature, pH 7, 10 mM Mg^2+^, 10 mM P_i_), a mixture of ATP and ADP will feature a ratio of 1 ATP to 10 million ADPs [59,81]. The chemiosmotic free-energy-converting engine drives the ATP/ADP ratio to between 10 and 1000 ATPs per 1 ADP, that is, roughly 9 orders of magnitude away from where it wants to be [59]. It is the Gibbs free energy available in this disequilibrium which drives virtually all, by themselves strongly endergonic, cellular reactions “uphill”, that is, towards the disequilibria (i.e., entropy decrease) characterizing life. The fact that it is the ATP/ADP couple (or, more rarely, the ADP/AMP or even the ATP/AMP couples) which makes metabolic reactions proceed towards the right (for life, but wrong for thermodynamics) direction led to the somewhat unfortunate, since misleading, notion of ATP as the cellular energy “currency”. For example, a one molar solution of ATP provides zero kiloJoules if it is in the presence of a 10^7^-fold excess of ADP while its energy content is −57 kJ/mol^−1^ if ADP is present at 10^−3^ molar concentration [59,60]. As a matter of example, one cannot pay for a single biosynthetic conversion of pyruvate into tyrosine by inserting an ATP-coin into the money slot of the involved enzymes but it is the compounded “weight” of the ATP mountain wanting to reverse into an ADP mountain that pushes the bulk of the precursor molecules all the way up to their thermodynamically highly unlikely product, tyrosine [21,58].

Noteworthily, the free energy that can be extracted from out-of-equilibrium ATP/ADP (or ADP/AMP) couples has next to nothing to do with the adenosine-moiety since equilibrium constants determined for the adenosine-devoid, i.e., purely “inorganic”, triphosphate/diphosphate (also called pyrophosphate) or pyrophosphate/phosphate couples lie in the same range as those of their adenosine-linked counterparts [81]. Consequently, certain ATP-driven enzymes also perform their tasks in the presence of non-equilibrium polyphosphates [82,83].

### 5.3. Chemiosmosis, a Physical Phenomenon in Biological Disguise

Chemiosmosis is fundamentally based on the presence of a topological compartment (containing roughly 1 to a few femtoliters) sealed from the environment by a dielectric barrier enabling the whole system to work like a capacitor (Figure 4). This barrier is erroneously considered (and even described as such in most biochemistry textbooks) as “the lipid bilayer membrane”. CryoEM images obtained on native biological membranes have shown that the surfaces of these membranes are in fact by a large margin dominated by proteins and not by lipids [84,85,86,87], a fact already known to quantitative biochemists half a century ago [88]. The low dielectric constant of the interior space of proteins (ε ≈ 4) indeed makes them perfect constituents of the type of dielectric barrier encountered in and required by chemiosmotic systems (high dielectric-constant environments, such as water with its ε of about 80, strongly reduce electric fields via polarization-induced shielding of charges; the internal dielectric constant of proteins is commensurate with that of a pure lipid membrane allowing strong electric fields to be generated already by small charge asymmetries).

We therefore consider the commonly held view that first cellular entities were necessarily bounded by lipid-based membranes as conjecture and generally at odds with what we see in extant life. The paradigm of the lipid-bilayer membrane as the confinement separating the cellular interior (the “cytoplasm”) from the environment naturally led to numerous speculations in the origin-of-life field on how first lipids may have been synthesized prebiotically and how first micellar and then vesicular structures may have been formed [89,90,91]. We contend (based on the biological facts mentioned above) that the presence of lipids is not a strict prerequisite for the formation of (chemiosmotically functional) compartments but that the respective dielectric barriers may also have consisted purely of proteinaceous matter very much as is the case in extant gas vesicles [92] or bacterial microcompartments (BMCs) [93].

This dielectric barrier harbours electron transfer chains made up primarily from a variable number (according to the specific redox substrates used) of metalloproteins assisted in most but not all cases by a few small redox active organics (quinones, flavins, NAD etc). While electrons flow through these “bioenergetic chains”, an electrostatic and cation-concentration gradient (The term chemiosmosis is essentially a misnomer, since, contrary to the definition of osmosis, it doesn’t imply movement of water molecules but rather of cations (protons or sodium ions) to restore thermodynamic equilibrium conditions. However, the term is so firmly established in the biological literature that we will stick to it throughout this article. Nevertheless, true to Confucius’ insight that “The beginning of wisdom is to call things by their right name”, we would urge the bioenergetics community to try and come up with a more fitting term) is built up over the dielectric barrier, partly as a direct result of opposite barrier-sidedness of oxidation/reduction reactions and partly due to active vectorial transport of small cations such as protons or, more rarely, sodium ions described in more detail previously [31,74]. These gradients generate a cation-motive force over the roughly 5 nm-wide dielectric barrier which pushes the mentioned cations back through dedicated free-energy-converting machineries within the dielectric barrier towards the inside of the compartment. Within these machineries, the movement of the cations is coupled to phosphorylation reactions, displacing the ratios of in general ATP/ADP, but also PP_i_/P_i_ [94] far away from their equilibrium values (see previous paragraph).

The mechanism of chemiosmosis as summarized in Figure 4 looks eerily non-biological and predominantly involves physical/electrochemical principles (capacitor-like boundaries, redox gradients, electric fields, inorganic metal-clusters, inorganic phosphates and polyphosphates). Rather than being a highly evolved biological process, the core of chemiosmosis resembles a basic electrochemical device which is conceivable in the absence of any life-like features. What is more, it is a direct manifestation of how the 2nd law authorizes localized entropy decrease by taking its encompassing, larger, system closer to equilibrium (Figure 4, colour-coded correspondence between elements of chemiosmosis and individual terms of the 2nd law).

## 6. Free Energy Conversion Prior to Chemiosmosis?

In the above paragraphs, we have tried to impress on the reader the extraordinarily “abiotic” features of the chemiosmotic principle, the free-energy-converting mechanism of extant life. It therefore is far from obvious that chemiosmosis is, as commonly assumed, the high-end mechanism having required extended evolutionary timescales to appear and prior to which other, likely simpler but unknown, types of energy conversion must have existed. Rather, we contend that the chemiosmotic principle may have operated right at the beginning of living things and may in fact have been the very fundamental reason why life was able to appear in the first place.

Here we have to admit that “contend” isn’t “know”. There is, of course, no stringent argument excluding the frequently proposed series of pre-chemiosmosis types of free energy conversion. However, neither are there arguments necessarily requiring such processes! The empirical approach to elucidating the evolutionary pathway of free energy converting mechanisms is based on comparative biology and phylogenetic analyses. This approach allows to reach back to the last universal common ancestor (LUCA) of all living beings on Earth but no further! And all available evidence presently indicates that LUCA used chemiosmosis. This implies that putative pre-chemiosmotic processes, as intellectually appealing and intuitive as they may come, can never be proven to have indeed existed. Applying Occam’s razor, we therefore consider that the only truly empirical approach to elucidating the emergence of life-enabling free energy conversion is to try and draft scenarios which can rationalize a prebiotic appearance of the chemiosmotic mechanism. If this approach succeeds, we will have a plausible, thermodynamically viable and empirically supported, narrative for life’s emergence. If it doesn’t, a nearly infinite number of alternative scenarios can be thought of but we will never be able to tell whether they have anything to do with the actual evolutionary history of life on Earth.

## 7. Towards Scenarios Featuring Prebiotic Chemiosmotic Processes

The string of arguments developed so far in this article calls for emergence of life hypotheses based on the prebiotic presence of a chemiosmotic system of free energy conversion resembling the principle shown in Figure 4. While there are potentially a number of possible settings fulfilling this requirement, historically only the alkaline hydrothermal vent theory (AVT) so far took this necessity seriously. In fact, the original version of the AVT scenario [32,95] was strongly boosted by the realization that several aspects of chemiosmosis come as a natural consequence of the topology of the porous hydrothermal mounds and of the chemical and electrochemical properties of both the alkaline vent fluid and the surrounding ocean waters [68,69,70,71,96,97].

Several details of this newer version of the alkaline hydrothermal vent hypothesis [98] drew fierce criticism from both the bioenergetics [99] and the thermodynamics [22] communities. More recent versions of this hypothesis strive to address these well-taken points of criticism (a more detailed account of the “evolutionary” trajectory of the AVT from its emergence to the present day will be presented in Russell and Nitschke (“in preparation”)).

While the authors of the present article attempt to design and experimentally test scenarios generating close-to-lifelike manifestations of chemiosmotic processes in the framework of the AVT scenarios, we cannot and will not exclude the possibility that chemiosmosis may have arisen in other settings. However, the argumentation detailed throughout this article to our mind leaves no alternative to searching for a chemiosmotic emergence of life if both Schrödinger’s strictures and the experimental evidence for extant life’s universal way to comply with the 2nd law are taken seriously.

## 8. The Other “What Is Life” Question

Schrödinger’s insight that the 2nd law of thermodynamics necessarily implies that the generation of order in living matter must somehow be driven by environmental disequilibria, put the emergence of life on par with that of other dissipative-structure-forming phenomena of the physical world. However, extant life undeniably differs from such physical dissipative structures by the property of (error-prone) coded reproduction exposing it to Darwinian natural selection and thereby enabling evolutionary adaptation and innovation. It is this property which allowed life to persist as a dissipative structure through 4 billion years of evolution, to explore (and continue to do so) the available space of redox-disequilibrium-offering “habitats” on our planet and to eventually manage to reproduce the required redox tensions by using photon energy from our parent star [100]. Mutability together with natural selection thus accomplished the extraordinary feat of adding to Earth-bound, geochemical, sources of free energy a truly extraterrestrial one through hooking chemiosmosis up to light energy from the sun [101]. Such mutability may already have been present in replication processes based on peptides, that is, before the RNA-mediated coding systems had appeared, as worked out during the last two decades [102,103,104,105] with (again) metal-ions (e.g., Zn) playing crucial roles in recognition between template- and daughter-chains of peptides [104].

These considerations shine light on the fact that we are still sorely lacking a commonly accepted definition for the meaning of (the term) life. Since it is coded reproduction that sets life apart from other dissipative structures, considering life to only commence when coded reproduction set in appears perfectly defendable to us. Would that imply that all the considerations laid out in this article actually have no bearing on the emergence of life? We firmly argue to the contrary and in doing so we have Schrödinger on our side. All processes involved in putatively leading to the emergence of coded replication ineluctably entail a reduction in the entropy of the system and therefore have to be driven by a mechanism (in our minds likely chemiosmosis) capable of paying for this entropy decrease with free energy from the environment. The abiotic existence of such a free energy converting system therefore is the sine-qua-non condition for life to emerge.

## Figures and Tables

**Figure 1 life-14-00607-f001:**
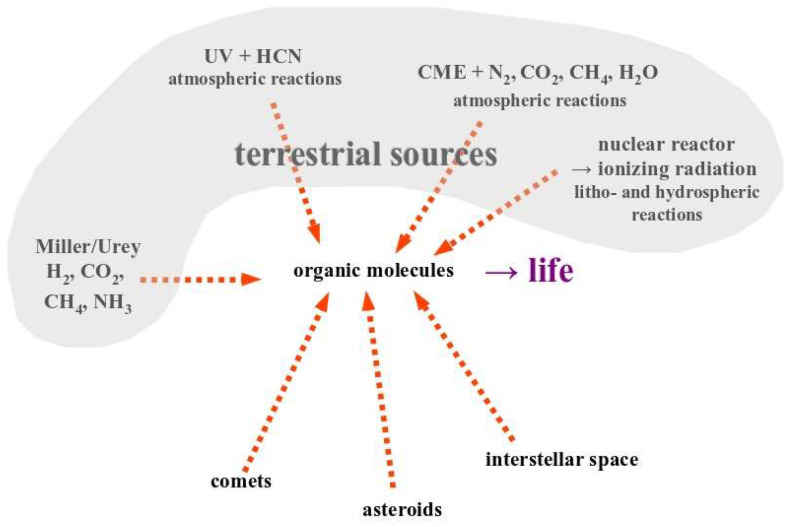
Diverse derivations of the prevailing paradigm for how life came into being on planet Earth. As discussed in the text, this paradigm holds that the presence of an ensemble of organic molecules is the necessary and sufficient condition for life to appear. When the first abiotic synthesis pathway for organics was discovered by Stanley Miller in the 1950s (left side of the figure), the question of life’s origin was considered fundamentally solved. In the meantime, a plethora of different ways to abiotically generate organics were found (only a few representatives of which are shown in this figure) and the detection of organic molecules in space even opened up the possibility that these “building blocks for life” may have been delivered from extraterrestrial sources. On the background of the mentioned paradigm, the debate presently mainly turns around the question of sorting out which specific pre-biotic synthesis pathway did indeed lead to life on Earth.

**Figure 2 life-14-00607-f002:**
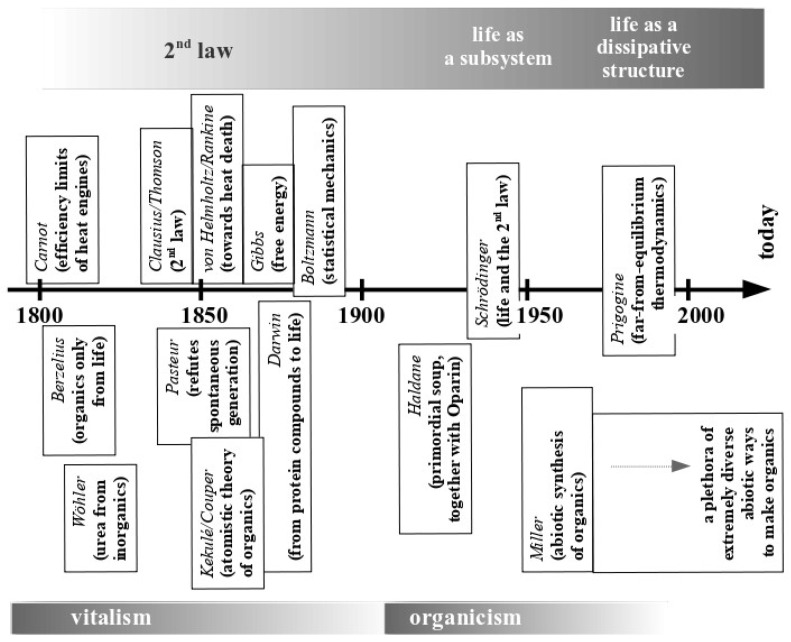
Timeline (from the beginning of the 19th century to the present day) of the development of the major concepts in the fields of thermodynamics (top part of the figure above the time-arrow) and in the biological/chemical sciences dealing with life and its birth on our planet (bottom part below the time-arrow). Up until the middle of the 20th century, these developments occurred with little or no cross-talk between the physical and the biological disciplines, reflecting the perceived incompatibilities of observed behaviours. Since Schrödinger’s venture into the life sciences [18] (although this topic was already touched upon by Boltzmann), the barrier between thermodynamics and biology was understood to be artificial later culminating in the rationalization of life as a far-from-equilibrium dissipative structure [19,20,21,22].

**Figure 3 life-14-00607-f003:**
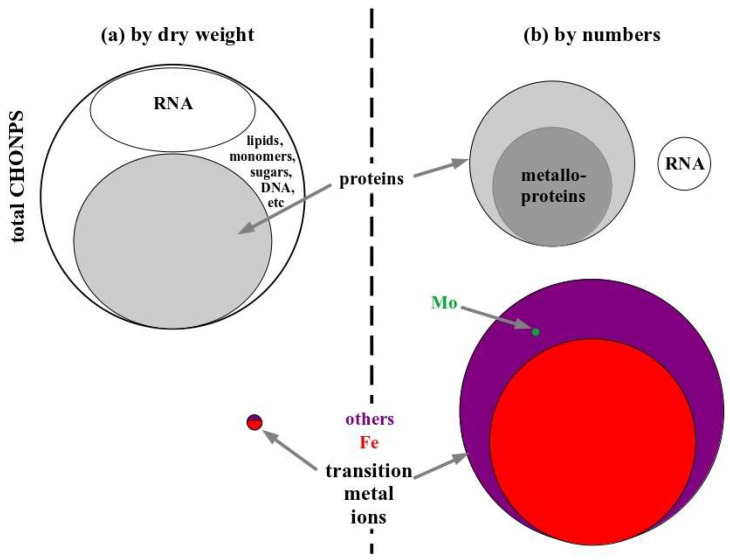
Comparison of the respective share of organic molecules and metal ions in the makeup of an *E. coli* cell when measured (**a**) by weight or (**b**) by numbers. The surface of the depicted ellipsoids is proportional to the relative proportions of the treated items.

**Figure 4 life-14-00607-f004:**
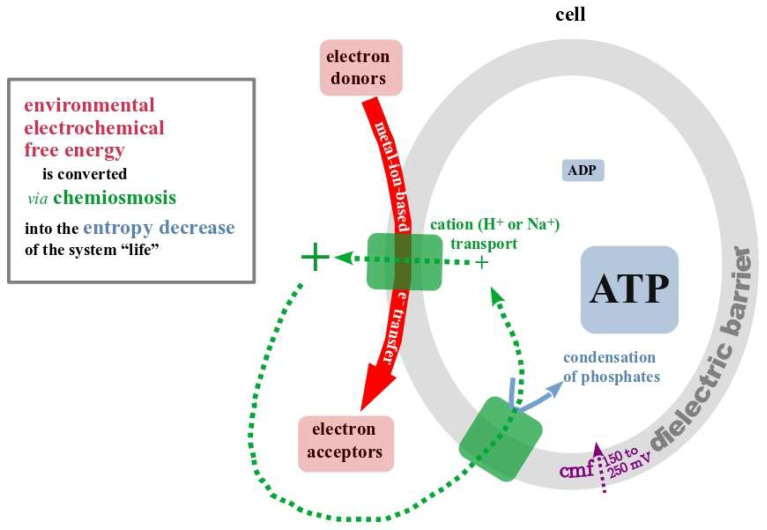
Schematic representation of life’s universal mechanism for converting environmental free energy into cellular entropy decrease, chemiosmosis. The correspondence between individual items in this process and terms of the 2nd law of thermodynamics (expressed in plain language on the left of the figure) is emphasized by colour-coding.

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
