# Peer review of "The Winding Road from Origin to Emergence (of Life)"

_life, 2024, doi:10.3390/life14050607_

Round 1

Reviewer 1 Report

Comments and Suggestions for Authors

The first thing I want to say is: thank you for writing such a clear and provocative paper! I truly enjoyed this review assignment more than almost any I can remember. While I disagree with some of the main conclusions, I still recommend acceptance of this paper in its current form.

That said, I do have a few suggestions and points of disagreement that the authors may (or may not) choose to act on.

1)        L. 73. I don’t understand the te modifier “so-called” in “so-called origin-of-life research.” 

2)        L. 144. It is overstating it to say that “Pasteur’s brilliant series of experiments DEFINITIVELY DISPROVED the millenia-old notion…. that life can spontaneously emerge from inanimate matter.” Certainly, many scientists were not convinced well into the 20th Century.

3)        L181. I suggest you reconsider the moniker “Organicism.” This is an established philosophical -ism: to quote wikipedia: “Organicism is the philosophical position that states that the universe and its various parts ought to be considered alive and naturally ordered, much like a living organism.” Instead, I would suggest organic-centrism.

4)        L. 203-. This section is a bit confusing. There can be lots of chance in evolution, thanks to the vaguaries of mutation, genetic drift, environmental change, and mass extinction. But that does not imply that the emergence of an evolvable chemical system in the first place was a matter of blind luck.

5)        L.285. The stars of your show are “FeNiMoWCo+” but you never explain why you picked these 5 actors over others such as Cu, Mn, and V…..  

6)        L.294-. Given how careful you are elsewhere to emphasize that what matters is disequilibrium, I am surprised that you did not discuss the most obvious argument for organic-centrism. If you look at the elemental composition of life and compare it to the planet as a whole, life is enriched in C H and N (and, to a lesser degree, P, K, F and Cl). Based on numbers I pulled together, the atomic fraction of living dry matter that is N is 240 times higher than the planet as a whole. C is enriched 125x, and H by 30x. In contrast, life is depleted in O, by a factor of about 3x, which aligns with the fact that the work of making life comes down to reducing carbon and nitrogen. All transitions metals are under-represented in life, iron, for example, by a factor of 10^-5. Given all this I think there is a good reason to consider the organic (redefined as CHNP) nature of life as very much one of the main things we need to explain! This is is consistent with your arguments that transition metal catalysis is very important, and energetics even more so, but suggests you are being somewhat unfair to organic chemistry.

7)        L483. I think this is a typo: “ATP/ADP ratio to between 10 and 1000 ATPs per 1 ATP” – last ATP should be ADP.

8)        L512. You might want to define dielectric and explain more clearly why: “..the interior space of proteins …makes them perfect constituents of the type of dielectric barrier…”

9)        L544. “in Fig. 4 looks eerily non-biological and predominantly involves physical/electrochemical principles” This seems disingenuous since you intentionally drew the figure to de-emphasize the role of organic molecules.

10)  Section 6 (and my critique of your model). You criticize most origin-of-life scientists for emphasizing one of the classes of matter that life uses, namely the organic one. However, but I think you make a similar mistake by emphasizing one particular biochemical process that life uses, namely chemiosmosis. I would propose that life is best thought of as a system that feeds on global and local disequilibria to make more of itself. Life requires autocatalysis and living entities can be seen as ecosystems of cooperating autocatalytic cycles (see https://royalsocietypublishing.org/doi/full/10.1098/rsif.2023.0346). Relevant to your argument, the cooperating cycles that compose life include both primarily organic ones (rTCA, glycoloysis, formose, etc.) AND inorganic ones (see: https://pubs.acs.org/doi/full/10.1021/jacs.3c07041). I personally think it is most parsimonious to imagine life emerging through the coupling of inorganic redox cycles with organic, carbon-fixing reducing ones.

This way of thinking also emphasizes a potential problem with your preferred hypothesis: A viable model in which the first barriers for chemiosmosis were inorganic would also have to show that this process was autocatalytic with respect to the barriers themselves such that creation of a cationic gradients resulted in the formation of more barrier molecules. Maybe this is part of your model, I don’t know, but based on this paper it seems that you might be focusing too much on one process of life (chemiosmosis) and ignoring the essential interdependence organic and inorganic autocatalytic processes.

Author Response

We will start by thanking the reviewer for his/her detailed and thoughtful evaluation of our ms and in particular for accepting that we might in places have an opinion on the emergence of life that differs from his/hers.

Point by point response:

1. We agree and we have cut the “so-called”. This qualifier actually was a left-over from a previous version of the ms wherein we addressed the difference in meaning between “origin” and “emergence” of life early on in the text.

2. The reviewer is perfectly right. We changed the text from “definitively disproved” to “severely challenged” (lines 145-146).

3. While the term “organicism” has the philosophical meaning as indicated by the reviewer, it has other meanings in different disciplines. I looked up the wikipedia site mentioned by the reviewer and found indeed the phrase cited by the reviewer at the top of this page. However, scrolling down the same page, wikipedia details the meaning of the term in other disciplines, such as “politics and sociology” or … “biology”. In this subsection, the term is explained exactly as we use it in our ms. To avoid confusion, we have now added a reference (new ref [24], Needham 1928 “Organicism in biology”) which details the meaning of the term as envisaged by the early proponents of the primordial soup scenario.

We emphasize that the term “organicism” is not at all synonymous with the “organic-centrism” proposed by the reviewer. Organicism does not refer to organic molecules but to organismal assemblies, which were considered by the early soup-people to afford the entropy-lowering abilities previously attributed to the vital force of organic molecules.

4. We insist that “chance” is different from “probability”! Yes, the effect of genetic drift, environmental change etc will have a non-deterministic, (i.e. chance-) effect on evolutionary trajectories, just as you might get any face up when you toss a dice once. However, as Monod had understood, imagining the fortuitous generation of an evolvable chemical system from an ensemble of building blocks is equivalent to getting the 1-dot-face upwards millions of times in a row, i.e. with a vanishing probability!

5. Very good point! We have now specified our choice in a new footnote 2, referred to in the text on line 318. The + in the FeNiMoWCo+ of course is an allusion to the LGBT+ terminology ;-)

6. We respectfully disagree with this line of arguments. Life almost certainly didn’t emerge from an environment featuring the average element abundances on planet Earth (and not even in the crust only) but in an aqueous environment likely heavily loaded with dissolved CO2 (the ancient atmosphere likely contained several bars of carbon dioxide) as well as nitrogen oxyanions and phosphates.

7. Well spotted. Thank you for pointing this out! It is corrected now.

8. Agreed. We have specified this now on lines 527-531.

9. We strongly disagree. All of section 5.3. as well as parts of section 4.2. indeed argue that transition metal ions rather than organics perform the relevant reactions. However, we also emphasize that the dielectric barrier necessarily is a result of organics (lipids or proteins with a preference for the latter). The organics therefore are an integral part of our putative primordial chemiosmotic system but they are a passive element (allowing the electric field to build up)! This is what we mean when we say that chemiosmosis looks eerily non-biological (in the sense where “biological” commonly – but erroneously, we would hold – is taken as “performed by organic molecules”). It is in fact performed by inorganic agents but needs the dielectric barrier, likely at life’s emergence formed from organic goo produced beforehand via mineral-mediated CO2-reduction or CH4-oxidation. And the chemiosmotic principle doubtlessly “predominantly involves physical/electrochemical principles”. We might add that this non-biological aspect of chemiosmosis was the very reason why the vast majority of the bioenergetics community initially vehemently rejected Peter Mitchell’s rationalization, leading to what is known as the “Oxphos-wars”.

10. Here we disagree fundamentally with the reviewer’s vision of what is essential for life to emerge. The autocatalytic cycles have initially been proposed to bootstrap macroscopic (organics-based) structures out of nano-scale reaction schemes. However, this process has since been recognized to only produce vast amounts of organic slime (Steven Benner’s “tar-problem”), i.e. an exponential entropy increase, rather than the defining feature of life, that is, generation of order. So, no! The carbon-fixing etc reactions cannot be on the same footing as the free-energy converting process; the latter must necessarily pre-date the former (and by the way, carbon fixation has been shown to also be performed by minerals). This fact is now acknowledged by many colleagues from the primordial (organic) soup community (see for example the articles by Pross and Pascal cited in our manuscript). The specifics of our proposal are merely that we make the case for present-day’s free energy converting mechanism (chemiosmosis) to go back all the way to the emergence of life rather than being a more recent invention following in evolutionary times other types of free energy conversion (that is, order generation).

Reviewer 2 Report

Comments and Suggestions for Authors

I like this article very much. Especially the importance of cofactors (including metals and metal complexes) in both the origin of as well as current life cannot be stressed often enough in my opinion. Similarly, life as a self-organizing emergent phenomenon (or dissipative structure as the authors refer to it), rather than a linear pathway to some basic building block, still seems to be too much underappreciated. Although it is of course important to know how the basic building blocks for life arose, as the authors rightly point out there seems to be too much emphasis on that part alone as *the* origin of life, whereas the question of their spontaneous organization into a self-sustaining complex system is still too often ignored. In that sense the jab at the RNA world hypothesis could have been more elaborate. But on the other hand, the authors are probably right that it doesn't really deserve much more attention anyway.

In short, I would love to see this article published as is!

Author Response

The only point of criticism which we could detect in this very-pleasant-to-read evaluation of our manuscript was that we could have taken on the RNA-world hypothesis in a “more elaborate” way. We in principle agree that the RNA-world deserves to be taken apart on the basis of thermodynamics and we were indeed tempted to do just that. However, while writing this piece, we felt that the (necessarily lengthy) RNA-part takes the drift of the manuscript too far from its main focus. We therefore decided to leave dealing with the RNA-world to a separate, future manuscript and only address the metal-ion-cofactor aspects of ribozymes.

Reviewer 3 Report

Comments and Suggestions for Authors

Keywords are missing?

In my modest opinion, the authors make a very worthy reading of what life is, from the point of view of essential contributions that have occurred since Chemistry is considered a science. There is little I can add, while it is impossible for me not to say anything about it. And my comments are in this sense, based on several decades dedicated to Bioinorganic Chemistry.

Yes, Chemistry is a young science, 200 years old, while the origin of life, and consequently Biological Chemistry, is estimated at almost 5 eons. It seems clear that life arises in aquatic (thermal) environments that favor appropriate molecular recognition, to which are added a host of favorable circumstances to give way to entities capable of two crucial things: Being able to reproduce and adapt. And this occurs, with astonishing frequency, reaching systems far from equilibrium, self-controlled by enzymes... Adaptation made it possible to overcome the transition from anoxygenic photosynthesis (which generates S as a by-product) to oxigenic photosynthesis (which generates dioxygen), condemning anaerobic beings to their almost total extinction. Another crucial issue is that the first step of genetic expression is (almost without exception) deposited in the function of the zinc fingers. If the first step is not taken, the next ones cannot be taken. And Zn is an abundant, bioavailable element lacking redox chemistry (which leaves it outside of the invention of chlorophyll).

Without going into the recent contributions of genetic engineering, the linking of Zn to genetic expression leads to the adaptation of living beings, as a source of biodiversity, the greatest guarantor of survival.

Author Response

We thoroughly concur with the reviewer’s opinion that the potential for adaptation of living beings, rooted in error-prone replication, is the crucial property distinguishing life from other dissipative structures as emphasized in section 8. of our manuscript. We also admit that Zn may have played a role in these replication processes and we have added respective information as well as a few relevant references in section 8. (lines 633-636 and refs [103-106]). Since Zn is redox-inactive, as mentioned by the reviewer, it is not expected to have played an active role (although possibly a structure-stabilizing one) in the free energy converting reactions which are the main topic of our manuscript.